# The Development of a Multi-Modal Cancer Rehabilitation (Including Prehabilitation) Service in Sheffield, UK: Designing the Active Together Service

**DOI:** 10.3390/healthcare12070742

**Published:** 2024-03-29

**Authors:** Liam Humphreys, Anna Myers, Gabriella Frith, Michael Thelwell, Katie Pickering, Gary H. Mills, Karen Kerr, Patricia Fisher, John Kidder, Carol Keen, Suzanne Hodson, Gail Phillips, Rachel Smith, Laura Evans, Sarah Thornton, Emma Dale, Louise Maxwell, Diana M. Greenfield, Robert Copeland

**Affiliations:** 1Academy of Sport and Physical Activity, Sheffield Hallam University, Sheffield S1 1WB, UK; a.myers@shu.ac.uk (A.M.); g.frith@shu.ac.uk (G.F.); m.thelwell@shu.ac.uk (M.T.); katie.pickering@shu.ac.uk (K.P.); 2Advanced Well-Being Research Centre, Sheffield Hallam University, Sheffield S1 1WB, UK; jhnkidder@aol.com (J.K.); g.phillips@shu.ac.uk (G.P.); r.j.copeland@shu.ac.uk (R.C.); 3Critical Care Directorate, Sheffield Teaching Hospitals NHS Foundation Trust, University of Sheffield, Sheffield S1 1WB, UK; g.h.mills@sheffield.ac.uk (G.H.M.);; 4Specialised Cancer Services, Sheffield Teaching Hospitals NHS Foundation Trust, Sheffield S10 2JF, UK; patriciafisher@nhs.net (P.F.); diana.greenfield@nhs.net (D.M.G.); 5Therapeutics and Palliative Care Directorate, Combined Community and Acute Care Group, Sheffield Teaching Hospitals NHS Foundation Trust, Sheffield S10 2JF, UK; carol.keen@nhs.net (C.K.); suzanne.hodson@nhs.net (S.H.); rachel.smith162@nhs.net (R.S.); laura.evans48@nhs.net (L.E.); 6Dietetic Service, Sheffield Teaching Hospitals NHS Foundation Trust, Sheffield S10 2JF, UK; sarah.thornton14@nhs.net; 7Department of Psychological Services, Sheffield Teaching Hospitals NHS Foundation Trust, Sheffield S10 2JF, UK; emma.dale5@nhs.net; 8Sheffield Teaching Hospitals NHS Foundation Trust, Sheffield S10 2JF, UK; louisemaxwell@nhs.net; 9Department of Oncology and Metabolism, University of Sheffield, Medical School Beech Hill Road, Sheffield S10 2RX, UK

**Keywords:** exercise prescription, exercise, physical activity, psychology, nutrition, cancer, oncology, rehabilitation, prehabilitation, barriers

## Abstract

Cancer patients undergoing major interventions face numerous challenges, including the adverse effects of cancer and the side effects of treatment. Cancer rehabilitation is vital in ensuring cancer patients have the support they need to maximise treatment outcomes and minimise treatment-related side effects and symptoms. The Active Together service is a multi-modal rehabilitation service designed to address critical support gaps for cancer patients. The service is located and provided in Sheffield, UK, an area with higher cancer incidence and mortality rates than the national average. The service aligns with local and regional cancer care objectives and aims to improve the clinical and quality-of-life outcomes of cancer patients by using lifestyle behaviour-change techniques to address their physical, nutritional, and psychological needs. This paper describes the design and initial implementation of the Active Together service, highlighting its potential to support and benefit cancer patients.

## 1. Introduction

Cancer rehabilitation (including prehabilitation) is central to high-quality cancer care [1]. It enables patients to prepare well for treatment, maximise outcomes from treatment, and minimise treatment-related side effects and symptoms [1], leading to enhanced quality of life. The demand for rehabilitation services will continue to grow as the population ages and more people are diagnosed with and surviving cancer [2]. As cancer transitions from a terminal disease to a long-term health condition [3], there is a need for lifestyle support through cancer rehabilitation services (including prehabilitation) to be integrated into the cancer care continuum [4].

Within the growing population of cancer survivors, two significant health concerns exist. The first is the concern regarding cancer recurrence and mortality. The second is the persistent adverse effects of cancer and its treatment [5]. There is increasing evidence that physical activity and psychological and nutritional states are related to prognosis and survival after cancer [6]. Observational studies have indicated that insufficient physical activity and poor nutrition are associated with disease-related outcomes such as recurrence, death from cancer, and overall mortality [7,8]. Evidence now highlights that regular exercise and a healthy diet before, during, and after treatment decreases the length of stay and the severity of treatment-related adverse side effects; additionally, it is associated with a reduced risk of cancer recurrence and comorbid conditions such as cardiovascular disease and diabetes [9]. Meeting the physical activity guidelines following a cancer diagnosis is associated with a 21–35% lower risk of cancer recurrence, a 28–44% reduced risk of cancer-specific mortality, and a 25–48% reduced risk of all-cause mortality [10].

### 1.1. Multi-Modal Cancer Rehabilitation

Rehabilitation was cited as a central element of cancer care and a vital theme of the Achieving World-Class Outcomes Strategy [11]. The National Health Service (NHS) Long Term Plan states the need to provide cancer patients with personalised care, including needs assessment, a care plan, and health and well-being information and support [12]. However, rehabilitation (including prehabilitation) is not routinely offered to cancer patients. 

Numerous attempts have been made to define “cancer rehabilitation” [13]. Herbert Dietz classified cancer rehabilitation into four categories that align with the different stages of the cancer journey: (1) preventative, (2) restorative, (3) supportive, and (4) palliative [14]. More recently, Silver and colleagues defined cancer rehabilitation as “medical care that should be integrated throughout the oncology care continuum and delivered by trained rehabilitation professionals who have it within their scope of practice to diagnose and treat patients’ physical, psychological, and cognitive impairments to maintain or restore function, reduce symptom burden, maximise independence, and improve quality of life in this medically complex population” [13].

The volume of cancer rehabilitation studies has grown substantially over the last two decades [15]. However, the availability of rehabilitation services for cancer patients remains minimal in many countries [1,16,17]. Only 1 in 200 cancer patients is estimated to have access to cancer rehabilitation services [18]. Just 9% of hospitals in the United Kingdom, 6% in Canada, 19% in Australia, and 1% of hospitals in the United States offer exercise-based cancer-rehabilitation services [18].

One of the critical barriers to the routine provision of cancer rehabilitation is the lack of funding; where funding is available, it is often short-term and reliant on charitable organisations [1]. Multi-centre trials [19] and services [20] evaluating prehabilitation and rehabilitation have begun in the UK. These programmes aim to assess cancer rehabilitation on a large scale as part of accepted clinical practice in the NHS [20]. This paper describes the development of a multi-modal cancer rehabilitation (including prehabilitation) service in Sheffield, UK and the approach to embedding it within routine clinical practice.

### 1.2. The Cancer Landscape in Sheffield

South Yorkshire (Sheffield, Rotherham, Barnsley, and Doncaster) has a catchment population of approximately 1.4 million people, with over 580,000 residing in Sheffield. There are over 8000 new cancer diagnoses per year in South Yorkshire, and in Sheffield alone, 3038 new diagnoses per year [21]. South Yorkshire has a higher incidence (624 new cancer diagnoses for every 100,000 people per year) [21] of cancer and higher cancer mortality rates (270 people in every 100,000 die from cancer each year) than the national average [21], with similar one-year survival rates (71.5% compared to the national average of 72.3%) [22]. Sheffield has one of only four dedicated cancer centres in England with a world-leading reputation as a centre of excellence in cancer treatment, care, and pioneering services for patients with late-treatment consequences. Patients across South Yorkshire and North Derbyshire receive oncological care from Weston Park Cancer Centre, part of Sheffield Teaching Hospitals (the primary provider of acute cancer services for the South Yorkshire region). Cancer services and pathways are organised through the local Cancer Alliance, part of the recently established South Yorkshire (SY) Integrated Care Board (ICB). The ICB aims to integrate the provision of health services across different organisations, including hospitals and community-based services, physical and mental health, and health and social care [23]. The SY and Bassetlaw (SYB) Cancer Alliance is one of 19 alliances across England. The SYB Cancer Alliance is responsible for coordinating and delivering key national and regional cancer priorities as part of the ICB. These priorities arose from the Cancer Taskforce’s recommendations [11] and were drawn from health inequalities data for the geographical population. Despite the infrastructure in SYB, historically, people with a cancer diagnosis had no equitable access to rehabilitation services. The Active Together service aims to address this critical support gap while aligning with the local and regional cancer care objectives of the SYB Cancer Alliance.

## 2. Development of the Active Together Service 

The Active Together service has been co-produced by cancer patients, clinical and professional stakeholders, and academics over a two-year period. The service has also been informed by our previous work in cancer rehabilitation [24], where we identified the critical factors needed to create a model of exercise support for cancer patients [25,26]. These include time during consultations to discuss rehabilitation, sufficient knowledge of the benefits of rehabilitation for cancer patients, and trust In the services that professionals refer patients to [25,26].

The programme of co-production involved meaningful and extensive consultation with patients, health professionals, professional leaders, commissioners, and academic experts to consider what a rehabilitation (including prehabilitation) service could look like, how to shape it to meet the contextual needs of Sheffield, and what elements are crucial in delivering an impactful and sustainable service. 

In total, 40 professionals supported the design process. Patient and professional groups and leaders communicated that the critical components of rehabilitation services included accessibility, safety, location, expertise, and capacity. The engagement of clinical teams across tumour groups was also deemed essential to foster buy-in from the programme’s start. One key finding was the range of expertise required to deliver an effective multi-modal rehabilitation (including prehabilitation) programme. However, there were insufficient levels of allied health professional (AHP) capacity within existing teams (i.e., physiotherapists, occupational therapists, and dietitians) to deliver a high-quality service. This was also the case for clinical psychologists and exercise professionals. Therefore, additional staff would need to be funded, recruited, and trained. 

Patients should be at the centre of any discussions about their care [27], and so we ensured meaningful engagement was prioritised with patients throughout the design process. We held informal patient-discussion groups to understand patients’ needs and preferences. Patients are also an integral part of our service-steering group. We held co-design sessions where prompts and service prototypes were used to tease out key service-design features. We also created patient personas using insight from patient narratives to help guide service design. Patient personas are fictional representations of different types of patients who may interact with a particular service or healthcare intervention [28]. Each persona included demographic information, medical history, psychosocial factors, preferences, and specific needs or goals related to the service or intervention being designed. Personas create a shared and persistent view of the potential user, which can be referred to when making design decisions [28]. The personas were particularly valuable in ensuring the service did not exacerbate existing inequalities. In total, 18 cancer patients supported the design process. Through these discussions, we gained valuable insights into the diverse needs, preferences, and priorities of cancer patients, which informed the content and the delivery style of the service discussed below. 

### 2.1. The Content of the Active Together Service

Active Together is an evidence-based, person-centred, multi-modal rehabilitation (including prehabilitation) service for people with a cancer diagnosis. Active Together provides support that spans the rehabilitation continuum and covers preventive rehabilitation (prehabilitation) following diagnosis, maintenance rehabilitation (during treatment), restorative rehabilitation (immediately post-treatment), and supportive rehabilitation (after treatment and discharge). The service aims to improve patient, clinical, and patient-reported outcomes such as fatigue, physical fitness before surgery, treatment-related side effects, treatment tolerance, postoperative complications, survival rates, and quality of life. 

### 2.2. Referral

All patients with a primary diagnosis of lung, colorectal, or upper gastrointestinal (UGI) cancer, aged ≥ 18 years and scheduled for curative treatment in Sheffield, were referred to Active Together following diagnosis. Lung, colorectal, and UGI cancer were chosen due to their prevalence within the Yorkshire region and the significant challenges they pose in terms of rehabilitation and supportive care needs. By focusing on these specific cancer types, we sought to address the unique physical, psychological, and social needs of patients undergoing treatment with these cancer types. Referrals are completed by HCPs (i.e., oncologists, cancer nurse specialists, and AHPs) from cancer services at STH. HCPs have previously identified the time taken to complete referral processes for services as a potential barrier to engagement [25]. Therefore, the Active Together service’s referral process is deliberately simple, making the process easier for HCPs and ensuring the patient can access support quickly. Once a referral has been made, a member of the Active Together team contacts the patient by telephone within 48 h and books the patient an initial assessment with a physiotherapist. This appointment will occur within seven days to both maximise the prehabilitation window and lower the risk of patient disengagement. Behaviour-change principles underpin the initial-contact telephone call with the patient and follow an ‘ask, advise, assist’ structure [29]. The consultation provides an opportunity to explore with the patient their attitude, motivation, and understanding of the benefits of engaging in the service. It is also an opportunity to empower patient choice, enhancing patient control to enable patient action [29].

### 2.3. The Initial Patient Needs Assessment

Patients attend an initial assessment with a specialist physiotherapist at Sheffield Hallam University’s Advanced Wellbeing Research Centre (AWRC) or at combined health and physical activity centres in Sheffield (i.e., the Move More Centres discussed below). During this initial assessment, each patient completes physical, nutritional, and psychological screening assessments to determine the level of support required in each area. A personalised care plan is then offered per the universal, targeted, and specialised model advocated in the prehabilitation guidance [30]. Patients may have different levels of need for each element (i.e., physical, nutritional, and psychological).

A screening algorithm is used to determine the patient’s level of need for each rehabilitation element. The screening algorithm was developed through consultation with professionals in each area (physiotherapists, dietitians, and clinical psychologists) and academics. Additionally, recommendations from the existing literature on the screening process were incorporated, such as appropriate screening tools and how to determine the level of intervention required [6,30]. Table 1 shows the measures used to determine the patient’s level of need in each area. Each measure is entered into the screening algorithm, which determines the patient’s need using a traffic-light system with green indicating low complexity, amber indicating moderate complexity, and red indicating high complexity (Table 2). The patient’s level of need identified during this initial assessment determines the support they will receive and who delivers it. The patient’s need is reassessed at key time points throughout the patient’s treatment pathway, and the support provided is adjusted accordingly.

### 2.4. Active Together Service Delivery Style and Behaviour-Change Approach

Initiating and maintaining a healthier lifestyle is complicated, and patients often require support to succeed [31]. The quality of this support can be enhanced by utilising behaviour-change theory as part of the design of a complex intervention [32]. Despite this, interventions in the cancer rehabilitation context rarely adequately describe the content and approach to behaviour change [33], with some exceptions [34,35]. The behavioural-change content of the Active Together programme, including the style of intervention delivery, has been developed using the behaviour-change wheel [36] to target specific behaviours and identify the most appropriate behaviour-change techniques (BCT’s) to support lifestyle change amongst people with a cancer diagnosis.

As a key part of the behaviour-change wheel, intervention functions were chosen to drive the desired changes [36]. The Active Together service includes six intervention functions: education, training, persuasion, enablement, modelling, and environmental restructuring. These functions aim to empower patients and provide them with the necessary tools and support [36]. For example, education is provided to patients about the benefits of regular exercise, nutrition, and psychological well-being. Training is provided to equip patients with the skills and knowledge necessary to perform the exercises effectively and safely. Enablement ensures patients can access necessary resources, such as exercise equipment or supportive programmes, to facilitate behaviour change.

After determining the appropriate intervention functions, the next step was selecting specific behaviour-change techniques (BCTs) aligned with the chosen functions. The Behaviour Change Taxonomy [37] provides a comprehensive list of 93 BCTs, allowing the Active Together service to tailor its approach to suit patients’ individual needs. For example, previous studies have demonstrated the importance of promoting self-efficacy as part of cancer recovery [38], which might include the promotion of mastery of behaviours such as exercise [39], setting achievable goals [40], and the inclusion of graded tasks, helping promote lifestyle change [35]. Two key behaviour-change techniques employed within our service are the utilisation of credible sources and the facilitation of social support. Firstly, by ensuring that our service is delivered by trained healthcare professionals, including physiotherapists, dietitians, psychologists, and specialist fitness instructors, we aim to provide patients with credible and trustworthy information and guidance. This approach not only enhances the reliability of the support provided but also instils confidence and reassurance in patients regarding the efficacy of the information and support they receive. Secondly, we recognise the importance of social support in fostering behaviour change and improving psychosocial well-being among cancer survivors. Our service integrates various strategies to facilitate social support, including group-based support, peer support, and the involvement of family members. By creating a supportive environment, we empower patients to make positive lifestyle changes and navigate the challenges of cancer and its treatment with resilience and strength. Table 3 shows the multi-modal BCTs used in the Active Together service aligned with the intervention functions, with examples describing how they are applied within the service. 

All patient interactions within the Active Together service are (i) delivered in a motivational interviewing (MI) style [41], adopting the principles of collaboration, compassion, evocation, and acceptance (known as the spirit of MI), and (ii) incorporate the “What Matters to You?” approach, a method of implementing patient-centred care and shared decision-making [42]. Communicating with patients in this way aims to promote patient empowerment and a sense of control. Practitioners who deliver the Active Together service have received comprehensive training to deliver the intervention and associated behaviour-change techniques, with competencies assessed via a process evaluation described in a forthcoming publication.

### 2.5. Phases of the Rehabilitation Continuum

Phase 1: Prehabilitation

Cancer prehabilitation is a process on the continuum of care that occurs between the time of cancer diagnosis and the beginning of acute treatment [43]. Patients enter the Active Together service prehabilitation phase after the initial needs assessment. The duration of the prehabilitation phase will depend on how soon after diagnosis the patient enters the service and how long after diagnosis the patient begins their acute cancer treatment. As a result, the degree to which a patient’s physical and psychological health can be improved before the start of treatment will vary between individuals. Even in cases where the prehabilitation phase is very short (≤2 weeks), it is hoped that the advice and support provided during the prehabilitation phase will bring about changes in behaviour that can be continued during and after treatment, enabling people to live well beyond their treatment [30].

To reduce the risk of patients contracting COVID-19, adaptations to programmes and procedures are made in line with NHS COVID-19 and infection-control policies at the time. Patients are also given guidance and nutritional supplements to support them from a dietetic perspective, as well as access to paper-based or online materials and regular phone calls with Active Together staff to support their psychological well-being.

Phase 2: Maintenance rehabilitation

During their acute treatment, patients enter the maintenance phase of Active Together. Although evidence suggests that physical activity is safe and feasible during cancer treatment [31], participation has been shown to reduce significantly [44]. During the maintenance phase, the service aims to support patients in maintaining their current physical, nutritional, and psychological status as much as possible, supporting them to cope with any distress caused by cancer and its treatment whilst minimising the impact of any treatment-related side effects [45]. The duration of the maintenance phase will depend on the type and duration of the treatment patients receive. Each individual receives a personalised exercise plan that can be completed in various settings depending on patient preference (the gym, a park or open space, or at home). Active Together will maintain contact with them via telephone, text, or email, depending on patient preference.

Phase 3: Restorative rehabilitation

Following the patient’s acute clinical treatment, they enter the restorative rehabilitation phase of the Active Together pathway. The restorative phase aims to support the patient in achieving optimal function following acute treatment [46]. At the start of the restorative phase, the patient repeats their initial needs assessment. The outcome of this review determines the patient’s prescribed exercise programme, nutrition plan, and psychological/behavioural support strategies. The restorative rehabilitation phase can last up to 12 weeks following the end of the cancer treatment, depending on individual patient needs. The emphasis in the restorative phase is on helping individuals return to their pre-treatment levels of health and well-being. This is achieved by setting functional goals involving strength and fitness within the context of what matters most to the patient (e.g., returning to work or being able to play with their grandchildren).

Phase 4: Supportive rehabilitation

The final phase of the Active Together programme is supportive rehabilitation, which aims to empower patients to achieve optimal health and well-being while living with long-term health conditions [47]. The supportive phase can last up to 12 weeks, depending on individual patient needs, and might involve referring the individual to existing community exercise schemes such as walking groups. This phase culminates with a final assessment of outcomes, with the goal that the person will have surpassed their pre-diagnosis fitness levels and can self-manage their physical, nutritional, and psychological well-being.

### 2.6. Active Together Workforce

Rehabilitation (including prehabilitation) guidance from Macmillan Cancer Support states that rehabilitation should be delivered by a multidisciplinary team [30]. Led by academics at Sheffield Hallam University and supported by professional leaders and clinicians at STH, funding was secured from Yorkshire Cancer Research to recruit the core Active Together team. This team consists of a consultant AHP, two physiotherapists, a dietitian, a clinical psychologist, two researchers with expertise in cancer exercise prescription and health behaviour change, two fitness instructors, a project manager, and an administrator. The team also includes a sustainability manager whose role is to explore the long-term future of the service (thee SYB Cancer Alliance funded this role). Figure 1 shows the staffing structure of the Active Together service.

While the Active Together service strives to offer comprehensive care, we acknowledge the limitation of not currently including occupational therapy within our multidisciplinary team. As highlighted in the World Health Organization’s Rehabilitation 2030 initiative, occupational therapy is integral to addressing functional limitations, promoting independence, and facilitating meaningful engagement in daily activities for individuals affected by cancer [48]. The Active Together service refers patients to occupational health services should they require support in that domain.

A clinical advisory group (CAG) was established to support Active Together’s design, development, and ongoing implementation. The purpose of the CAG is to provide clinical leadership and governance oversight to ensure the successful delivery of Active Together. The responsibilities of the CAG include addressing service issues and risks and resolving them. The CAG comprises clinical leads from each cancer pathway; medical specialists from cancer services, including anaesthetists, surgeons, oncologists, senior nursing, and AHP leads; academic leads for the programme; and patient representatives.

### 2.7. Service Delivery Setting

The programme-development stakeholder-discussion groups stressed that any service must be delivered outside of the hospital setting. Delivering the service outside of hospital settings allows for improved convenience and accessibility (e.g., car parking facilities). As part of the city-wide ‘Move More’ strategy [49], which adopts a whole system approach to increasing the population’s physical activity, Sheffield has three Move More centres that co-locate health-care services with exercise opportunities as part of a patient’s recovery and rehabilitation [50,51]. These centres currently deliver 100,000 clinical appointments per year, and the AWRC is a fourth centre taking the capacity up to 120,000. The AWRC and the three Move More centres include NHS clinical consulting rooms, education and training facilities, and exercise facilities within a leisure and sports facility. Located across the city in communities of high economic disadvantage, the three centres enable the co-location of patients, researchers, clinicians, allied health professionals, and sports- and exercise-medicine specialists, providing an ideal hub-and-spoke model to deliver an equitable service for patients across the city [50].

## 3. Next Steps

Implementation of Active Together began in January 2022, and an implementation phase of the service was undertaken to set up, test, and refine processes and procedures between February and June 2022. Implementing the Active Together service within the NHS presented unique advantages, such as access to a well-established infrastructure and a diverse patient population. However, challenges also arose, including operating in a multi-organisation collaboration, the complexity of the service, workforce demands, and expanding and scaling the service [52]. Since completing the implementation phase in June 2022, the service has been running at full capacity. A robust process and outcome evaluation has been embedded into the service [53], and the service will be continually refined based on the results. In addition to the ongoing evaluation and refinement of the Active Together service, the next steps include plans to scale up the service across South Yorkshire and expand its reach to include a broader range of cancer types. This expansion aligns with our commitment to providing equitable access to comprehensive cancer rehabilitation services for all individuals affected by cancer in our region. We will work closely with stakeholders, including healthcare providers, community organisations, and patient advocacy groups, to facilitate the implementation and integration of the service into existing cancer care pathways. Through strategic partnerships and collaboration, we aim to ensure the sustainability and effectiveness of the Active Together service as it continues to evolve and meet the evolving needs of our diverse patient population.

## 4. Conclusions

Cancer rehabilitation (including prehabilitation) is vital in ensuring patients have the support they need to maximise treatment outcomes and side effects associated with cancer and its treatment. Although there is growing evidence of the importance of rehabilitation, the availability of services across England is sporadic [1]. We identified an unmet need for cancer rehabilitation (including prehabilitation) in Sheffield and are addressing this by implementing the Active Together service. The service was co-designed using insight from previous research [25], service delivery, and robust stakeholder engagement, which prioritised the voice of patients. Robust evaluation is now needed to evidence the service’s acceptability and effectiveness.

## Figures and Tables

**Figure 1 healthcare-12-00742-f001:**
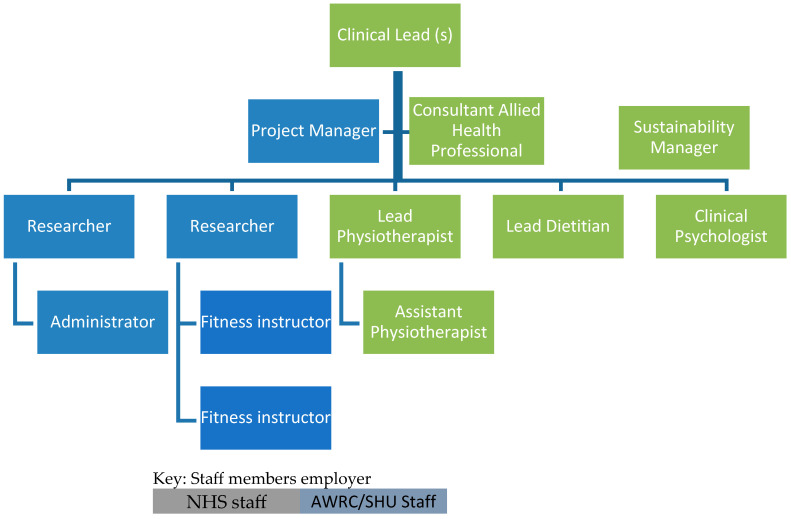
Active Together staffing structure.

**Table 1 healthcare-12-00742-t001:** Measures used to determine the patient’s level of need.

Rehabilitation Discipline	Measure
Physical activity needs criteria	Age Six-minute walk test (metres walked) History of any cardiac or metabolic complication that would impact physical activity Severe hypertension > 200 mm Hg and/or a diastolic blood pressure of >110 mm Hg Any other significant comorbidity or condition which, in the opinion of a medical professional, should be considered within the physical activity risk stratification that could compromise the safety of the patient
Nutritional needs criteria	Percentage weight change (past 1 and 6 months). Current food intake Symptoms preventing normal food intake Activities and function Any other significant comorbidity or condition which, in the opinion of a medical professional, should be considered within the nutritional risk stratification that could compromise the safety of the patient
Psychological needs criteria	Patient Health Questionnaire (PHQ-9) Generalised Anxiety Disorder (GAD-7) Current diagnosed mental health condition/s Previous mental health condition/s Any other significant factor which, in the opinion of a medical professional, should be considered within the psychological needs assessment that could compromise the safety of the patient

**Table 2 healthcare-12-00742-t002:** Patient level of need matrix and corresponding level of support with physical activity, nutrition, and psychological well-being.

Patient Need (Support Provided)	Support for Each Level of Need (What?, Who?, Where?)
Low complexity/need (Universal) The patient is healthy, with no additional physical, nutritional, and/or psychological risk factors and requires no specialist support.	What?
	Physical activity Home-based independent exercise Community-based facilities Nutrition Signposting to dietary management tools Psychological Alleviate distress from cancer and prepare the patient psychologically for treatment by signposting to psychological management tools.
	Who?
	Physical activity Self-guided exercise Technology supported Researchers Nutrition Provided written information provided by cancer teams Psychological Self-referral to Improving Access to Psychological Therapies (IAPT), for example, online or group
	Where?
	Individual preference of the patient
Moderate complexity/need (Targeted) The patient has some physical, nutritional, and/or psychological risk factors and requires monitoring and some specialist support and monitoring.	What?
	Physical activity Group or supervised exercise programme 2 × per week Nutrition Referred to generic NHS dietitian support Group-based nutrition support sessions Psychological Alleviate distress from cancer and prepare patient psychologically for treatment
	Who?
	Physical activity Cancer Exercise Specialist instructors with guidance from physiotherapists Nutrition Some dietitian support Psychological Psychological support via workshops and self-directed learning
	Where?
	AWRC fitness suite/clinic rooms Some home-based
High complexity/need (Specialist) The patient has significant physical, nutritional, and/or psychological risk factors and requires frequent specialist support.	What?
	Physical activity Physiotherapist Supervised exercise programme 2 × per week Nutrition Access to dietitian 1 × per week via telephone or face-face Psychological Alleviate distress from cancer and prepare patients psychologically for treatment
	Who?
	Physical activity Physiotherapist Support from a cancer exercise specialist Nutrition Specialist cancer dietitian support Psychological One-to-one psychological support with a researcherand signposting to additional support if needed. Workshops can be attended as long as patients are psychologically stable.
	Where?
	AWRC fitness suite AWRC clinic rooms

**Table 3 healthcare-12-00742-t003:** Multi-modal behaviour-change techniques used in Active Together aligned with intervention functions.

Intervention Function	Behaviour-Change Technique (BCT No. (BCTTv1))	Example of Application within the Multi-Modal Active Together Service
Education	Information about emotional consequences (5.6)	All staff delivering the service are trained to support patient psychological needs.
	Feedback on behaviour (2.2)	Fitness instructors or physiotherapists provide feedback on behaviour during exercise sessions.
	Self-monitoring of behaviour (2.3)	Patients are encouraged to keep a food diary where they record their daily meals and snacks, enabling them to track their eating patterns and identify areas for improvement.
	Reduce negative emotions (11.2)	Patients are advised and helped to develop strategies for ways to reduce negative emotions.
	Information about health consequences (5.1)	Participants are informed of the benefits of enhancing their fitness, optimising nutrition, and managing mental well-being prior to cancer treatment.
Persuasion	Credible source (9.1)	Trained professionals deliver all aspects of the service (see Section 2.6 Active Together Workforce).
	Information about emotional consequences (5.6)	Patients attend a workshop on managing psychological health. All staff delivering the service are trained to support patient psychological needs.
	Feedback on behaviour (2.2)	Dieticians provide feedback to the patient on changes made to diet and eating behaviour.
	Verbal persuasion and capability (15.1)	Fitness instructors provide verbal support and encouragement during exercise sessions.
Training	Demonstration of the behaviour (6.1)	Fitness instructors/physiotherapists demonstrate the correct exercise techniques during sessions.
	Instruction on how to perform a behaviour (4.1)	A dietitian instructs patients on correct nutritional balance (e.g., eat well plate) and provides step-by-step instructions on how to read food labels to make informed choices.
	Feedback on behaviour (2.2)	Fitness instructors or physiotherapists provide feedback on behaviour during exercise sessions.
	Self-monitoring of behaviour (2.3)	Heart-rate monitors and RPE are used to measure exercise intensity during supervised sessions.
	Graded tasks (8.7)	Exercise sessions are progressed during individual sessions. The exercise dose is increased towards 150 min per week.
	Behavioural practice/rehearsal (8.1)	Dieticians and patients practice portion control by using visual aids or portion plates to understand appropriate serving sizes.
	Biofeedback (2.6)	Heart-rate monitors and wearables are used to provide patients with feedback on exercise intensity during supervised and independent sessions.
Environmental restructuring	Adding objects to the environment (12.5)	Patients can access heart-rate monitors, wearables, logbooks, and resistance bands to facilitate unsupervised sessions.
	Reduce negative emotions (11.2)	Patients are advised to develop strategies for ways to reduce negative emotions.
Modelling	Demonstration of the behaviour (6.1)	Demonstrating relaxation techniques in psychological workshops.
Enablement	Goal setting (behaviour) (1.1)	The goal of the exercise programme is for patients to exercise 150 min per week.
	Goal setting (outcome) (1.3)	Collaboratively setting outcome-related goals, such as achieving or maintaining a certain weight or body mass index (BMI), to monitor progress and motivate patients.
	Adding objects to the environment (12.5)	Patients can access heart-rate monitors, wearables, logbooks, and resistance bands to facilitate unsupervised sessions.
	Problem-solving (1.2)	The patient is prompted to identify barriers and solutions to lifestyle change with the Active Together team (physiotherapist, dietitian, and fitness instructor).
	Action planning (1.4)	Helping patients create specific action plans for implementing healthy eating habits during busy workdays or social events.
	Self-monitoring of behaviour (2.3)	Patients are encouraged to keep a food diary where they record their daily meals and snacks, enabling them to track their eating patterns and identify areas for improvement.
	Generalisation of target behaviour (8.6)	Patients are advised to apply exercise, nutritional, and psychological techniques they have learnt at home.
	Review behaviour goal(s) (1.5)	Fitness instructors/physiotherapists/dieticians/psychologists review patient’s progress in achieving behaviour-related goals, providing support, and adjusting strategies as needed.
	Review outcome goal(s) (1.7)	Fitness instructors/physiotherapists/dieticians/psychologists discuss with patients how they are working towards their goals. Decisions are made on whether to update the goal or create a new goal.
	Social support (emotional) (3.3)	Patients have access to supervised group-based exercise sessions.

## Data Availability

Data are contained within the article.

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
