# Peer review of "The Development of a Multi-Modal Cancer Rehabilitation (Including Prehabilitation) Service in Sheffield, UK: Designing the Active Together Service"

_healthcare, 2024, doi:10.3390/healthcare12070742_

Round 1
Reviewer 1 Report
Comments and Suggestions for Authors
This is a multi-modal project aiming at the unmet needs of the cancer population in the Sheffield UK area. I have some comments for this manuscript.
1. Thorough introduction to different team members and their roles. Detailed lists of Behavior Change Techniques vs. Examples of application within Active Together service are helpful for interested investigators who wish to replicate this project. 2. This project included patients with cancers of upper GI/colorectal/lung due to local epidemiology or any other considerations. Please address more concerns. 3. This project included the domains of nutrition, physical activity, and psychosocial activity. Most techniques implemented focused on individual improvement such as dietary control, exercises, relaxation skills ... etc. However, social participation was only mentioned in the "clinical advisory group". The actual social interaction in real life, which can reflect the daily functions and holistic health as well as quality of life was suggested to be approached and discussed by this paper. 4. The World Health Organization launched its new initiative in rehabilitation, Rehabilitation 2030, by bringing together lots of professionals to assist them in creating the vision. The occupational therapists can guide the patients in lifestyle arrangements to improve the overall health, wellness perception, and quality of life for cancer survivors and other diagnosed groups. The following references were advised to the authors to facilitate their discussion or further implementation.Richards, L. G., & Vall´ ee, C. (2020). Editorial—Not just mortality and morbidity but also function: Opportunities and challenges for occupational
therapy in the World Health Organization’s Rehabilitation 2030 initiative. American Journal of Occupational Therapy, 74, 7402070010. https://doi.org/
10.5014/ajot.2020.742005
Yang, S. Y., Wang, J. D., & Chang, J. H. (2020). Occupational therapy to improve quality of life for colorectal cancer survivors: A randomized clinical trial.
Supportive Care in Cancer, 28, 1503–1511. https://doi.org/10.1007/s00520-019-04971-2
Author Response
We thank the reviewer for their review and suggestions to improve our paper. We appreciate their time and effort. Below, we have detailed our responses to the suggestions. The reviewer comments are provided with our responses in bullet points. We believe we have addressed all comments and the manuscript is much improved as a result. We hope the revised manuscript is accepted for publication in your journal.
Thorough introduction to different team members and their roles. Detailed lists of Behavior Change Techniques vs. Examples of application within Active Together service are helpful for interested investigators who wish to replicate this project.
- We appreciate your positive feedback on the introduction. We aimed to provide clarity on the collaborative efforts involved in our research, and we are glad to hear that you found it helpful.
- Furthermore, we are pleased that you found the detailed lists of Behaviour Change Techniques helpful. Our intention was to offer a comprehensive framework for interested investigators who may seek to replicate or adapt our project.
This project included patients with cancers of upper GI/colorectal/lung due to local epidemiology or any other considerations. Please address more concerns.
- Thank you for your comment regarding the inclusion criteria for patients in our project. We acknowledge that the rationale behind our patient selection criteria could have been more explicitly stated in the manuscript.
- We have added the following to page 4, lines 163-167: Lung, colorectal, and UGI cancer were chosen due to their prevalence within the Yorkshire region and the significant challenges they pose in terms of rehabilitation and supportive care needs. By focusing on these specific cancer types, we sought to address the unique physical, psychological, and social needs of patients undergoing treatment with these cancer types.
This project included the domains of nutrition, physical activity, and psychosocial activity. Most techniques implemented focused on individual improvement such as dietary control, exercises, relaxation skills ... etc. However, social participation was only mentioned in the "clinical advisory group". The actual social interaction in real life, which can reflect the daily functions and holistic health as well as quality of life was suggested to be approached and discussed by this paper.
- We appreciate your comment regarding the inclusion of social participation as a crucial domain within our project. While our manuscript primarily focused on interventions such as exercise, dietary, and psychological support, social support is a crucial aspect of the service. Social support is included as a behaviour change technique in table 3.
- In response to your suggestion, we have included a discussion on the role of social support (232-244). We have also included a discussion of the behaviour change technique ‘credible source’, which we feel is important to discuss but was previously missing.
The World Health Organization launched its new initiative in rehabilitation, Rehabilitation 2030, by bringing together lots of professionals to assist them in creating the vision. The occupational therapists can guide the patients in lifestyle arrangements to improve the overall health, wellness perception, and quality of life for cancer survivors and other diagnosed groups. The following references were advised to the authors to facilitate their discussion or further implementation.
Richards, L. G., & Vall´ ee, C. (2020). Editorial—Not just mortality and morbidity but also function: Opportunities and challenges for occupational therapy in the World Health Organization’s Rehabilitation 2030 initiative. American Journal of Occupational Therapy, 74, 7402070010. https://doi.org/10.5014/ajot.2020.742005
Yang, S. Y., Wang, J. D., & Chang, J. H. (2020). Occupational therapy to improve quality of life for colorectal cancer survivors: A randomized clinical trial.
Supportive Care in Cancer, 28, 1503–1511. https://doi.org/10.1007/s00520-019-04971-2
- Thank you for your comment regarding the potential inclusion of occupational therapy within our cancer rehabilitation service.
- While our current service does not explicitly incorporate occupational therapy, we recognise the significance of this discipline in comprehensive rehabilitation care. We agree that occupational therapists bring unique expertise in addressing functional limitations, promoting independence, and facilitating meaningful engagement in daily activities for individuals affected by cancer.
- We have included the following section with reference to the WHO Rehabilitation 2030 (Section 2.6 page 14): While the Active Together service strives to offer comprehensive care, we acknowledge the limitation of not currently including occupational therapy within our multidisciplinary team. As highlighted in the World Health Organization's Rehabilitation 2030 initiative, occupational therapy is integral to addressing functional limitations, promoting independence, and facilitating meaningful engagement in daily activities for individuals affected by cancer (WHO, Rehabilitation 2030). The Active Together service refers patients to occupational therapy services should they require support in that domain.
Reviewer 2 Report
Comments and Suggestions for Authors
Overall this paper is very well written and covers an important topic, the logistics of operationalizing cancer rehabilitation (including prehabilitation) into cancer care in a large health care system.
Title: Fine
Abstract: Fine
Section 1. Introduction:Fine
Section 2. Lines 134-143 on page 5. Regarding the patient input on “what mattered most", what was specifically learned about this in the patient discussion groups? Preceding paragraphs contain some detailing of crucial components of rehabilitation services, but we don’t really learn about what is most crucial on the patient side, other than that “personas” were created. What are the different persona types?
Section 2.3. Initial patient needs assessment.
At one point do physicians become involved in the physical activity/rehabilitation process? Is physiatry available? Is this program completely independent of physician oversight and/or consultation? Do any of the Clinical Leads (Figure 1) have a rehabilitation or exercise science background?
Section 2.4. Beginning of last paragraph is missing content.
Section 2.7 Service delivery setting. Why did the stakeholder groups want the services to be delivered outside of the hospital setting? What else is important to stakeholder groups? (This is a redundant concern from previous point made about the beginning of Section 2.)
3. Pilot phase.
It is noted that 100% of patients who attended their initial assessment went on to participate in the program. Is there any data on level of adherence, or about which particular phase (ie prehab, during-treatment, survivorship) had the best participation?
Discussion needed??? A missing piece from the overall coverage has to do with the context of this program being implemented within England’s NHS. It would be interesting to hear from the authors regarding any particular advantages and/or challenges of operationalizing this program within that system and how these efforts might translate to other health care systems. Or, at least have a statement if it is beyond their scope to consider this.
Table 2. Moderate complexity section. What is a “Level 4” exercise instructor?
Table 2. High complexity section. Shouldn’t physician input be part of this section, at least in some cases?
Table 3. Please clarify and reference SMART goals.
References
Generally the references appear appropriate. Some may be missing information. For example, for reference #1 the exact source is unclear. Is this an NHS document? References #23 and #49 are also not contextualized. Reference #46 does not indicate the journal.
Author Response
We thank the reviewer for their review and suggestions to improve our paper. We appreciate their time and effort. Below, we have detailed our responses to the suggestions. The reviewer comments are provided with our responses in bullet points. We believe we have addressed all comments and the manuscript is much improved as a result. We hope the revised manuscript is accepted for publication in your journal.
Overall, this paper is very well written and covers an important topic, the logistics of operationalizing cancer rehabilitation (including prehabilitation) into cancer care in a large health care system.
- Thank you for your positive feedback on our manuscript. We are pleased to hear that you found the paper well-written and that it addresses an important topic in cancer care
Title: Fine
Abstract: Fine
Section 1. Introduction: Fine
Section 2. Lines 134-143 on page 5. Regarding the patient input on “what mattered most", what was specifically learned about this in the patient discussion groups? Preceding paragraphs contain some detailing of crucial components of rehabilitation services, but we don’t really learn about what is most crucial on the patient side, other than that “personas” were created. What are the different persona types?
- The service incorporates the "What Matters to You?" approach within our interactions with patients to ensure that their needs are effectively addressed. We have removed the mention of ‘what mattered most’ from the development section to avoid confusion.
- On Page 4 lines 137-138: We held informal patient discussion groups to understand patient's needs and preferences. Patients are also in our service steering group.
- On page 4 lines 149-151: Through these discussions, we gained valuable insights into the diverse needs, preferences, and priorities of cancer patients, which informed the content and the delivery style of the service discussed below.
- We have also included a broader description of patient personas On Page 4 lines 142-145: Patient personas are fictional representations of different types of patients who may interact with a particular service or healthcare intervention. Each persona included demographic information, medical history, psychosocial factors, preferences, and specific needs or goals related to the service or intervention being designed.
Section 2.3. Initial patient needs assessment.
At one point do physicians become involved in the physical activity/rehabilitation process? Is physiatry available? Is this program completely independent of physician oversight and/or consultation? Do any of the Clinical Leads (Figure 1) have a rehabilitation or exercise science background?
- In the Active Together service clinical professionals (oncologists, surgeons, consultants, anaesthetists, cancer nurse specialists) typically become involved in the physical activity/rehabilitation process at the point of a patient's cancer diagnosis. All patients with lung, colorectal or upper GI cancer are referred to Active Together by these healthcare providers.
- The Active Together clinical lead is a physiotherapist by background and has a wealth of rehabilitation knowledge and experience. On entering the service, all patients undergo a needs assessment to determine their physical and psychological state and tailor the support they are offered accordingly. The initial assessments are conducted by physiotherapists.
- The Active Together service has minimal contact with General Practitioners (GPs) primarily due to the service's integration into the cancer treatment pathway. As patients are referred to the service at the point of their cancer diagnosis, their care journey becomes predominantly centred around oncologists, specialists, and the multidisciplinary cancer care team. In cases where additional health concerns are identified during the initial assessment, such as uncontrolled hypertension, we adhere to a protocol of promptly referring the patient back to their general practitioner for further evaluation.
Section 2.4. Beginning of last paragraph is missing content.
- We cannot find where content is missing
Section 2.7 Service delivery setting. Why did the stakeholder groups want the services to be delivered outside of the hospital setting? What else is important to stakeholder groups? (This is a redundant concern from previous point made about the beginning of Section 2.)
- Page 15 added: Delivering the service outside of hospital settings allows for improved convenience and accessibility (e.g. car parking facilities).
- Pilot phase.
It is noted that 100% of patients who attended their initial assessment went on to participate in the program. Is there any data on the level of adherence or about which particular phase (e.g., prehab, during-treatment, survivorship) had the best participation?
- The full-service evaluation is ongoing. Following comments from reviewer 3, we have removed the pilot data section. The protocol for the evaluation is currently under review and the results of the evaluation will be published in due course.
Discussion needed??? A missing piece from the overall coverage has to do with the context of this program being implemented within England’s NHS. It would be interesting to hear from the authors regarding any particular advantages and/or challenges of operationalizing this program within that system and how these efforts might translate to other health care systems. Or, at least have a statement if it is beyond their scope to consider this.
- Page 16 added: Implementing the Active Together service within the NHS has presented unique advantages, such as access to a well-established infrastructure and a diverse patient population. The service also posed challenges related to operating in a multi-organisational collaboration, the complexity of the service, workforce demands, and expanding and scaling the service.
- The key learning from the set-up phase of the Active Together service are discussing elsewhere (Keen et al. 2023) and we have cited this paper in the text. https://www.mdpi.com/2227-9032/11/23/3007
Table 2. Moderate complexity section. What is a “Level 4” exercise instructor?
- Level 4 is a specialist qualification level for fitness/exercise instructors. We understand that this is ambiguous, so we have replaced with the full title - Cancer Exercise Specialist instructor.
Table 2. High complexity section. Shouldn’t physician input be part of this section, at least in some cases?
- We acknowledge the importance of physician input in managing patients with complex medical needs. While physician input is not explicitly detailed in the initial section of our service, we ensure that patients receive comprehensive care through collaboration with referring clinicians and the clinical expertise within our multidisciplinary team. Through the team's collective expertise, we can effectively address the diverse needs of patients, including those with high complexity.
- Additionally, we recognise the significance of medical considerations such as contraindications to exercise, which may necessitate direct physician involvement. In cases where such contraindications are identified during the initial assessment, such as uncontrolled hypertension, we adhere to a protocol of promptly referring the patient back to their general practitioner for further evaluation.
- Our interdisciplinary approach ensures that patients with high complexity receive the necessary support and interventions tailored to their individual needs.
Table 3. Please clarify and reference SMART goals.
- Reviewer 3 suggested a simplification of the content in Table 3. We have therefore removed the mention of SMART goals.
References
Generally the references appear appropriate. Some may be missing information. For example, for reference #1 the exact source is unclear. Is this an NHS document? References #23 and #49 are also not contextualized. Reference #46 does not indicate the journal.
- We thank the reviewer for highlighting the missing information in the references. These have now been updated.
Reviewer 3 Report
Comments and Suggestions for Authors
Thank you for the opportunity to review your manuscript. Well written, interesting novel project. I have a few areas to note:
The manuscript is mainly descriptive which offers the reader a clear understanding of the topic. However you provide a snapshot of findings from your pilot but this is not backed up with ethical approval or methodology and methods, research process etc. This could be presented in a further research paper.
Line 188 there is a bold statement inserted - Error
Line 221 appears part of sentence missing
Table 3 - far too detailed, difficult to understand and does not make sense considering the text that follows.
please explain whether the "patient" I would have thought this term would not be used considering the Active Together approach underpinning this programme so I will use the term person or individual's home environment is assessed as well as their family bringing together an holistic approach?
Numerous statements required by the journal need completing at the end of the manuscript.
Thank you
Author Response
We thank the reviewer for their review and suggestions to improve our paper. We appreciate their time and effort. Below, we have detailed our responses to the suggestions. The reviewer comments are provided with our responses in bullet points. We believe we have addressed all comments and the manuscript is much improved as a result. We hope the revised manuscript is accepted for publication in your journal.
The manuscript is mainly descriptive which offers the reader a clear understanding of the topic. However, you provide a snapshot of findings from your pilot, but this is not backed up with ethical approval or methodology and methods, research process, etc. This could be presented in a further research paper.
- On reflection, we agree that the pilot phase data would be better presented in a subsequent paper, so we have removed it and replaced it with a ‘next steps’ section. This section discusses the evaluation protocol (currently under review), roll out of the service, and the experiences of implementing the service.
Line 188 there is a bold statement inserted – Error
- Removed
Line 221 appears part of sentence missing
- I cannot find where content is missing
Table 3 - far too detailed, difficult to understand and does not make sense considering the text that follows.
- We have simplified the table by providing only one example per behaviour change technique
- The content in table 3 relates to the information discussed in section 2.4.
please explain whether the "patient" I would have thought this term would not be used considering the Active Together approach underpinning this programme so I will use the term person or individual's home environment is assessed as well as their family bringing together an holistic approach?
- Given that the Active Together service is embedded within routine clinical practice, we consider "patient" to be the appropriate term to use. The term "patient" accurately reflects the setting and the healthcare provider's perspective. We aim to ensure that our language is consistent with the clinical nature of the service while maintaining a person-centred approach in our care delivery.
Numerous statements required by the journal need completing at the end of the manuscript.
- Thank you, we have provided the information to the journal within the online system.
Round 2
Reviewer 1 Report
Comments and Suggestions for Authors
No further questions.
Author Response
We thank the reviewer for their further review and suggestions. Below we have provided our responses to the suggestions. The reviewer comments are provided with our responses in bullet points.
In the last response to Reviewer #2 you stated: "We thank the reviewer for highlighting the missing information in the references. These have now been updated". But I noticed that no updates appear to have been made to the reference list.
• Apologies, we have now added the changes (highlighted in yellow)
At line 201 there is still written "Error! Reference source not found".
• We believe this was due to an error in the cross-referencing of the tables in the text, which we have now removed. This should have resolved the problem.
At lines 249-251, it is not clear if "Table 1" at the beginning of the sentence was added or deleted.
• This has been added due to the previous deletion of a table (linked to the cross-referencing mentioned above). However, the font was different from the other text shown. This has been corrected.